# Backprop KF: Learning Discriminative Deterministic State Estimators

**Tuomas Haarnoja, Anurag Ajay, Sergey Levine, Pieter Abbeel**
{haarnoja, anuragajay, svlevine, pabbeel}@berkeley.edu
Department of Computer Science, University of California, Berkeley

## Abstract

Generative state estimators based on probabilistic filters and smoothers are one of the most popular classes of state estimators for robots and autonomous vehicles. However, generative models have limited capacity to handle rich sensory observations, such as camera images, since they must model the entire distribution over sensor readings. Discriminative models do not suffer from this limitation, but are typically more complex to train as latent variable models for state estimation. We present an alternative approach where the parameters of the latent state distribution are directly optimized as a deterministic computation graph, resulting in a simple and effective gradient descent algorithm for training discriminative state estimators. We show that this procedure can be used to train state estimators that use complex input, such as raw camera images, which must be processed using expressive nonlinear function approximators such as convolutional neural networks. Our model can be viewed as a type of recurrent neural network, and the connection to probabilistic filtering allows us to design a network architecture that is particularly well suited for state estimation. We evaluate our approach on synthetic tracking task with raw image inputs and on the visual odometry task in the KITTI dataset. The results show significant improvement over both standard generative approaches and regular recurrent neural networks.

## 1  Introduction

State estimation is an important component of mobile robotic applications, including autonomous driving and flight [22]. Generative state estimators based on probabilistic filters and smoothers are one of the most popular classes of state estimators. However, generative models are limited in their ability to handle rich observations, such as camera images, since they must model the full distribution over sensor readings. This makes it difficult to directly incorporate images, depth maps, and other high-dimensional observations. Instead, the most popular methods for vision-based state estimation (such as SLAM [22]) are based on domain knowledge and geometric principles. Discriminative models do not need to model the distribution over sensor readings, but are more complex to train for state estimation. Discriminative models such as CRFs [16] typically do not use latent variables, which means that training data must contain full state observations. Most real-world state estimation problem settings only provide partial labels. For example, we might observe noisy position readings from a GPS sensor and need to infer the corresponding velocities. While discriminative models can be augmented with latent state [18], this typically makes them harder to train.

We propose an efficient and scalable method for discriminative training of state estimators. Instead of performing inference in a probabilistic latent variable model, we instead construct a deterministic computation graph with equivalent representational power. This computation graph can then be optimized end-to-end with simple backpropagation and gradient descent methods. This corresponds to a type of recurrent neural network model, where the architecture of the network is informed by the

structure of the probabilistic state estimator. Aside from the simplicity of the training procedure, one of the key advantages of this approach is the ability to incorporate arbitrary nonlinear components into the observation and transition functions. For example, we can condition the transitions on raw camera images processed by multiple convolutional layers, which have been shown to be remarkably effective for interpreting camera images. The entire network, including the observation and transition functions, is trained end-to-end to optimize its performance on the state estimation task.

The main contribution of this work is to draw a connection between discriminative probabilistic state estimators and recurrent computation graphs, and thereby derive a new discriminative, deterministic state estimation method. From the point of view of probabilistic models, we propose a method for training expressive discriminative state estimators by reframing them as representationally equivalent deterministic models. From the point of view of recurrent neural networks, we propose an approach for designing neural network architectures that are well suited for state estimation, informed by successful probabilistic state estimation models. We evaluate our approach on a visual tracking problem, which requires processing raw images and handling severe occlusion, and on estimating vehicle pose from images in the KITTI dataset [8]. The results show significant improvement over both standard generative methods and standard recurrent neural networks.

## 2   Related Work

Some of the most successful methods for state estimation have been probabilistic generative state space models (SSMs) based on filtering and smoothing (Figure 1). Kalman filters are perhaps the best known state estimators, and can be extended to the case of nonlinear dynamics through linearization and the unscented transform. Non-parametric filtering methods, such as particle filtering, are also often used for tasks with multimodal posteriors. For a more complete review of state estimation, we refer the reader to standard references on this topic [22].

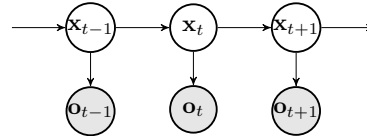

Figure 1: A generative state space model with hidden states $\mathbf{x}_i$ and observation $\mathbf{o}_t$ generated by the model. $\mathbf{o}_t$ are observed at both training and test time.

Generative models aim to estimate the distribution over state observation sequences $\mathbf{o}_{1:T}$ as originating from some underlying hidden state $\mathbf{x}_{1:T}$, which is typically taken to be the state space of the system. This becomes impractical when the observation space is extremely high dimensional, and when the observation is a complex, highly nonlinear function of the state, as in the case of vision-based state estimation, where $\mathbf{o}_t$ corresponds to an image viewed from a robot's on-board camera. The challenges of generative state space estimation can be mitigated by using complex observation models [14] or approximate inference [15], but building effective generative models of images remains a challenging open problem.

As an alternative to generative models, discriminative models such as conditional random fields (CRFs) can directly estimate $p(\mathbf{x}_t|\mathbf{o}_{1:t})$ [16]. A number of CRFs and conditional state space models (CSSMs) have been applied to state estimation [21, 20, 12, 17, 9], typically using a log-linear representation. More recently, discriminative fine-tuning of generative models with nonlinear neural network observations [6], as well as direct training of CRFs with neural network factors [7], have allowed for training of nonlinear discriminative models. However, such models have not been extensively applied to state estimation. Training CRFs and CSSMs typically requires access to true state labels, while generative models only require observations, which often makes them more convenient for physical systems where the true underlying state is unknown. Although CRFs have also been combined with latent states [18], the difficulty of CRF inference makes latent state CRF models difficult to train. Prior work has also proposed to optimize SSM parameters with respect to a discriminative loss [1]. In contrast to this work, our approach incorporates rich sensory observations, including images, and allows for training of highly expressive discriminative models.

Our method optimizes the state estimator as a deterministic computation graph, analogous to recurrent neural network (RNN) training. The use of recurrent neural networks (RNNs) for state estimation has been explored in several prior works [24, 4, 23, 19], but has generally been limited to simple tasks without complex sensory inputs such as images. Part of the reason for this is the difficulty of training general-purpose RNNs. Recently, innovative RNN architectures have been successful at mitigating this problem, through models such as the long short-term memory (LSTM) [10] and the

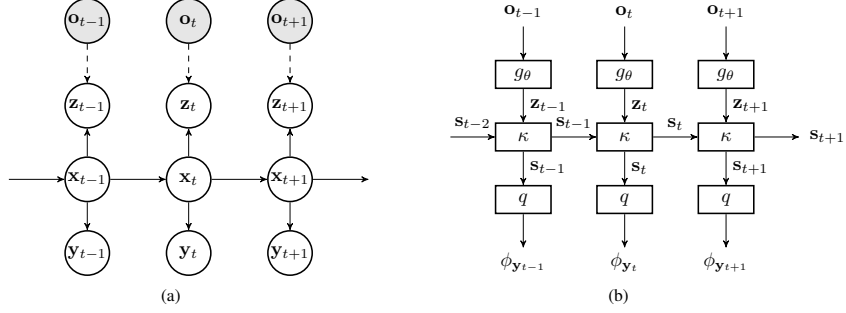

(a)                                                                (b)

Figure 2: (a) Standard two-step engineering approach for filtering with high-dimensional observations. The generative part has hidden state $\mathbf{x}_t$ and two observations, $\mathbf{y}_t$ and $\mathbf{z}_t$, where the latter observation is actually the output of a second deterministic model $\mathbf{z}_t = g_\theta(\mathbf{o}_t)$, denoted by dashed lines and trained explicitly to predict $\mathbf{z}_t$. (b) Computation graph that jointly optimizes both models in (a), consisting of the deterministic map $g_\theta$ and a deterministic filter that infers the hidden state given $\mathbf{z}_t$. By viewing the entire model as a single deterministic computation graph, it can be trained end-to-end using backpropagation as explained in Section 4.

gated recurrent unit (GRU) [5]. LSTMs have been combined with vision for perception tasks such as activity recognition [3]. However, in the domain of state estimation, such black-box models ignore the considerable domain knowledge that is available. By drawing a connection between filtering and recurrent networks, we can design recurrent computation graphs that are particularly well suited to state estimation and, as shown in our evaluation, can achieve improved performance over standard LSTM models.

## 3 Preliminaries

Performing state estimation with a generative model directly using high-dimensional observations $\mathbf{o}_t$, such as camera images, is very difficult, because these observations are typically produced by a complex and highly nonlinear process. However, in practice, a low-dimensional vector, $\mathbf{z}_t$, which can be extracted from $\mathbf{o}_t$, can fully capture the dependence of the observation on the underlying state of the system. Let $\mathbf{x}_t$ denote this state, and let $\mathbf{y}_t$ denote some labeling of the states that we wish to be able to infer from $\mathbf{o}_t$. For example, $\mathbf{o}_t$ might correspond to pairs of images from a camera on an automobile, $\mathbf{z}_t$ to its velocity, and $\mathbf{y}_t$ to the location of the vehicle. In that case, we can first train a discriminative model $g_\theta(\mathbf{o}_t)$ to predict $\mathbf{z}_t$ from $\mathbf{o}_t$ in feedforward manner, and then filter the predictions to output the desired state labels $\mathbf{y}_{1:t}$. For example, a Kalman filter with hidden state $\mathbf{x}_t$ could be trained to use the predicted $\mathbf{z}_t$ as observations, and then perform inference over $\mathbf{x}_t$ and $\mathbf{y}_t$ at test time. This standard approach for state estimation with high-dimensional observations is illustrated in Figure 2a.

While this method may be viewed as an engineering solution without a probabilistic interpretation, it has the advantage that $g_\theta(\mathbf{o}_t)$ is trained discriminatively, and the entire model is conditioned on $\mathbf{o}_t$, with $\mathbf{x}_t$ acting as an internal latent variable. This is why the model does not need to represent the distribution over observations explicitly. However, the function $g_\theta(\mathbf{o}_t)$ that maps the raw observations $\mathbf{o}_t$ to low-dimensional predictions $\mathbf{z}_t$ is not trained for optimal state estimation. Instead, it is trained to predict an intermediate variable $\mathbf{z}_t$ that can be readily integrated into the generative filter.

## 4 Discriminative Deterministic State Estimation

Our contribution is based on a generalized view of state estimation that subsumes the naïve, piecewise-trained models discussed in the previous section and allows them to be trained end-to-end using simple and scalable stochastic gradient descent methods. In the naïve approach, the observation function $g_\theta(\mathbf{o}_t)$ is trained to directly predict $\mathbf{z}_t$, since a standard generative filter model does not provide for a straightforward way to optimize $g_\theta(\mathbf{o}_t)$ with respect to the accuracy of the filter on the labels $\mathbf{y}_{1:T}$. However, the filter can be viewed as a computation graph unrolled through time, as shown in Figure 2b. In this graph, the filter has an internal state defined by the posterior over $\mathbf{x}_t$. For

example, in a Kalman filter with Gaussian posteriors, we can represent the internal state with the tuple $\mathbf{s}_t = (\boldsymbol{\mu}_{\mathbf{x}_t}, \boldsymbol{\Sigma}_{\mathbf{x}_t})$. In general, we will use $\mathbf{s}_t$ to refer to the state of any filter. We also augment this graph with an output function $q(\mathbf{s}_t) = \phi_{\mathbf{y}_t}$ that outputs the parameters of a distribution over labels $\mathbf{y}_t$. In the case of a Kalman filter, we would simply have $q(\mathbf{s}_t) = (\mathbf{C}_{\mathbf{y}} \boldsymbol{\mu}_{\mathbf{x}_t}, \mathbf{C}_{\mathbf{y}} \boldsymbol{\Sigma}_{\mathbf{x}_t} \mathbf{C}_{\mathbf{y}}^{\mathsf{T}})$, where the matrix $\mathbf{C}_{\mathbf{y}}$ defines a linear observation function from $\mathbf{x}_t$ to $\mathbf{y}_t$.

Viewing the filter as a computation graph in this way, $g_\theta(\mathbf{o}_t)$ can be trained discriminatively on the entire sequence, rather than individually on single time steps. Let $l(\phi_{\mathbf{y}_t})$ be a loss function on the output distribution of the computation graph, which might, for example, be given by $l(\phi_{\mathbf{y}_t}) = -\log p_{\phi_{\mathbf{y}_t}}(\mathbf{y}_t)$, where $p_{\phi_{\mathbf{y}_t}}$ is the distribution induced by the parameters $\phi_{\mathbf{y}_t}$, and $\mathbf{y}_t$ is the label. Let $\mathcal{L}(\theta) = \sum_t l(\phi_{\mathbf{y}_t})$ be the loss on an entire sequence with respect to $\theta$. Furthermore, let $\kappa(\mathbf{s}_t, \mathbf{z}_{t+1})$ denote the operation performed by the filter to compute $\mathbf{s}_{t+1}$ based on $\mathbf{s}_t$ and $\mathbf{z}_{t+1}$. We can compute the gradient of $l(\theta)$ with respect to the parameters $\theta$ by first recursively computing the gradient of the loss with respect to the filter state $\mathbf{s}_t$ from the back to the front according to the following recursion:

$$\frac{d\mathcal{L}}{d\mathbf{s}_{t-1}} = \frac{d\phi_{\mathbf{y}_{t-1}}}{d\mathbf{s}_t} \frac{d\mathcal{L}}{d\phi_{\mathbf{y}_{t-1}}} + \frac{d\mathbf{s}_t}{d\mathbf{s}_{t-1}} \frac{d\mathcal{L}}{d\mathbf{s}_t}, \tag{1}$$

and then applying the chain rule to obtain

$$\nabla_\theta \mathcal{L}(\theta) = \sum_{t=1}^{T} \frac{d\mathbf{z}_t}{d\theta} \frac{d\mathbf{s}_t}{d\mathbf{z}_t} \frac{d\mathcal{L}}{d\mathbf{s}_t}. \tag{2}$$

All of the derivatives in these equations can be obtained from $g_\theta(\mathbf{o}_t)$, $\kappa(\mathbf{s}_{t-1}, \mathbf{z}_t)$, $q(\mathbf{s}_t)$, and $l(\phi_{\mathbf{y}_t})$:

$$\frac{d\mathbf{s}_t}{d\mathbf{s}_{t-1}} = \nabla_{\mathbf{s}_{t-1}} \kappa(\mathbf{s}_{t-1}, \mathbf{z}_t), \qquad \frac{d\mathbf{s}_t}{d\mathbf{z}_t} = \nabla_{\mathbf{z}_t} \kappa(\mathbf{s}_{t-1}, \mathbf{z}_t),$$

$$\frac{d\mathcal{L}}{d\phi_{\mathbf{y}_t}} = \nabla_{\phi_{\mathbf{y}_t}} l(\phi_{\mathbf{y}_t}), \qquad \frac{d\phi_{\mathbf{y}_t}}{d\mathbf{s}_t} = \nabla_{\mathbf{s}_t} q(\mathbf{s}_t), \qquad \frac{d\mathbf{z}_t}{d\theta} = \nabla_\theta g_\theta(\mathbf{o}_t). \tag{3}$$

The parameters $\theta$ can be optimized with gradient descent using these gradients. This is an instance of backpropagation through time (BPTT), a well known algorithm for training recurrent neural networks.

Recognizing this connection between state-space models and recurrent neural networks allows us to extend this generic filtering architecture and explore the continuum of models between filters with a discriminatively trained observation model $g_\theta(\mathbf{o}_t)$ all the way to fully general recurrent neural networks. In our experimental evaluation, we use a standard Kalman filter update as $\kappa(\mathbf{s}_t, \mathbf{z}_{t+1})$, but we use a nonlinear convolutional neural network observation function $g_\theta(\mathbf{o}_t)$. We found that this provides a good trade-off between incorporating domain knowledge and end-to-end learning for the task of visual tracking and odometry, but other variants of this model could be explored in future work.

## 5   Experimental Evaluation

In this section, we compare our deterministic discriminatively trained state estimator with a set of alternative methods, including simple feedforward convolutional networks, piecewise-trained Kalman filter, and fully general LSTM models. We evaluate these models on two tasks that require processing of raw image input: synthetic task of tracking a red disk in the presence of clutter and severe occlusion; and the KITTI visual odometry task [8].

### 5.1   State Estimation Models

Our proposed model, which we call the "backprop Kalman filter" (BKF), is a computation graph made up of a Kalman filter (KF) and a feedforward convolutional neural network that distills the observation $\mathbf{o}_t$ into a low-dimensional signal $\mathbf{z}_t$, which serves as the observation for the KF. The neural network outputs both a point observation $\mathbf{z}_t$ and an observation covariance matrix $\mathbf{R}_t$. Since the network is trained together with the filter, it can learn to use the covariance matrix to communicate the desired degree of uncertainty about the observation, so as to maximize the accuracy of the final filter prediction.

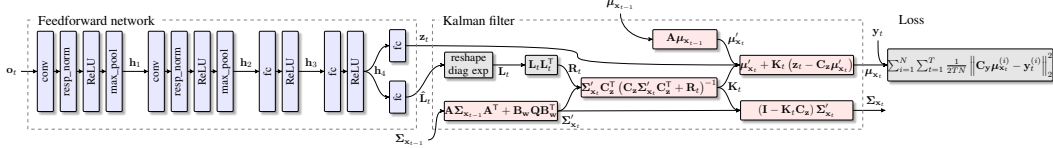

Figure 3: Illustration of the computation graph for the BKF. The graph is composed of a feedforward part, which processes the raw images $\mathbf{o}_t$ and outputs intermediate observations $\mathbf{z}_t$ and a matrix $\hat{\mathbf{L}}_t$ that is used to form a positive definite observation covariance matrix $\mathbf{R}_t$, and a recurrent part that integrates $\mathbf{z}_t$ through time to produce filtered state estimates. See Appendix A for details.

We compare the backprop KF to three alternative state estimators: the "feedforward model", the "piecewise KF", and the "LSTM model". The simplest of the models, the feedforward model, does not consider the temporal structure in the task at all, and consists only of a feedforward convolutional network that takes in the observations $\mathbf{o}_t$ and outputs a point estimate $\hat{\mathbf{y}}_t$ of the label $\mathbf{y}_t$. This approach is viable only if the label information can be directly inferred from $\mathbf{o}_t$, such as when tracking an object. On the other hand, tasks that require long term memory, such as visual odometry, cannot be solved with a plain feedforward network. The piecewise KF model corresponds to the simple generative approach described in Section 3, which combines the feedforward network with a Kalman filter that filters the network predictions $\mathbf{z}_t$ to produce a distribution over the state estimate $\hat{\mathbf{x}}_t$. The piecewise model is based on the same computation graph as the BKF, but does not optimize the filter and network together end-to-end, instead training the two pieces separately. The only difference between the two graphs is that the piecewise KF does not implement the additional pathway for propagating the uncertainty from the feedforward network into the filter, but instead, the filter needs to learn to handle the uncertainty in $\mathbf{z}_t$ independently. An example instantiation of BKF is depicted in Figure 3. A detailed overview of the computational blocks shown in the figure is deferred to Appendix A.

Finally, we compare to a recurrent neural network based on LSTM hidden units [10]. This model resembles the backprop KF, except that the filter portion of the graph is replaced with a generic LSTM layer. The LSTM model learns the dynamics from data, without incorporating the domain knowledge present in the KF.

## 5.2 Neural Network Design

A special aspect of our network design is a novel response normalization layer that is applied to the convolutional activations before applying the nonlinearity. The response normalization transforms the activations such that the activations of layer $i$ have always mean $\mu_i$ and variance $\sigma_i^2$ regardless of the input to the layer. The parameters $\mu_i$ and $\sigma_i^2$ are learned along with other parameters. This normalization is used in all of the convolutional networks in our evaluation, and resembles batch normalization [11] in its behavior. However, we found this approach to be substantially more effective for recurrent models that require backpropagation through time, compared to the more standard batch normalization approach, which is known to require additional care when applied to recurrent networks. It has been since proposed independently from our work in [2], which gives an in-depth analysis of the method. The normalization is followed by a rectified linear unit (ReLU) and a max pooling layer.

## 5.3 Synthetic Visual State Estimation Task

Our state estimation task is meant to reflect some of the typical challenges in visual state estimation: the need for long-term tracking to handle occlusions, the presence of noise, and the need to process raw pixel data. The task requires tracking a red disk from image observations, as shown in Figure 4. Distractor disks with random colors and radii are added into the scene to occlude the red disk, and the trajectories of all disks follow linear-Gaussian dynamics, with a linear spring force that pulls the disks toward the center of the frame and a drag force that prevents high velocities. The disks can temporally leave the frame since contacts are not modeled. Gaussian noise is added to perturb the motion. While these model parameters are assumed to be known in the design of the filter, it is a straightforward to learn also the model parameters. The difficulty of the task can be adjusted by increasing or decreasing the number of distractor disks, which affects the frequency of occlusions.

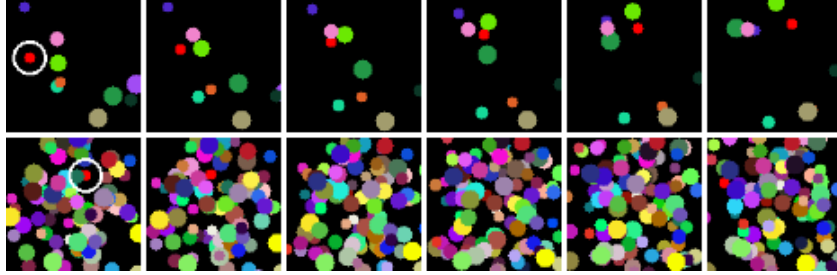

Figure 4: Illustration of six consecutive frames of two training sequences. The objective is to track the red disk (circled in the the first frame for illustrative purposes) throughout the 100-frame sequence. The distractor disks are sampled for each sequence at random and overlaid on top of the target disk. The upper row illustrates an easy sequence (9 distractors), while the bottom row is a sequence of high difficulty (99 distractors). Note that the target is very rarely visible in the hardest sequences.

Table 1: Benchmark Results

| State Estimation Model | # Parameters | RMS test error $\pm\sigma$ |
|---|---|---|
| feedforward model | 7394 | $0.2322 \pm 0.1316$ |
| piecewise KF | 7397 | $0.1160 \pm 0.0330$ |
| LSTM model (64 units) | 33506 | $0.1407 \pm 0.1154$ |
| LSTM model (128 units) | 92450 | $0.1423 \pm 0.1352$ |
| **BKF (ours)** | **7493** | $\mathbf{0.0537 \pm 0.1235}$ |

The easiest variants of the task are solvable with a feedforward estimator, while the hardest variants require long-term tracking through occlusion. To emphasize the sample efficiency of the models, we trained them using 100 randomly sampled sequences.

The results in Table 1 show that the BKF outperforms both the standard probabilistic KF-based estimators and the more powerful and expressive LSTM estimators. The tracking error of the simple feedforward model is significantly larger due to the occlusions, and the model tends to predict the mean coordinates when the target is occluded. The piecewise model performs better, but because the observation covariance is not conditioned on $\mathbf{o}_t$, the Kalman filter learns to use a very large observation covariance, which forces it to rely almost entirely on the dynamics model for predictions. On the other hand, since the BKF learns to output the observation covariances conditioned on $\mathbf{o}_t$ that optimize the performance of the filter, it is able to find a compromise between the observations and the dynamics model. Finally, although the LSTM model is the most general, it performs worse than the BKF, since it does not incorporate prior knowledge about the structure of the state estimation problem.

To test the robustness of the estimator to occlusions, we trained each model on a training set of 1000 sequences of varying amounts of clutter and occlusions. We then evaluated the models on several test sets, each corresponding to a different level of occlusion and clutter. The tracking error as the test set difficulty is varied is shown Figure 5. Note that even in the absence of distractors, BKF and LSTM models outperform the feedforward model, since the target occasionally leaves the field of view. The performance of the piecewise KF does not change significantly as the difficulty increases: due to the high amount of clutter during training, the piecewise KF learns to use a large observation covariance and rely primarily on feedforward estimates for prediction. The BKF achieves the lowest error in nearly all cases. At the same time, the BKF also has dramatically fewer parameters than the LSTM models, since the transitions correspond to simple Kalman filter updates.

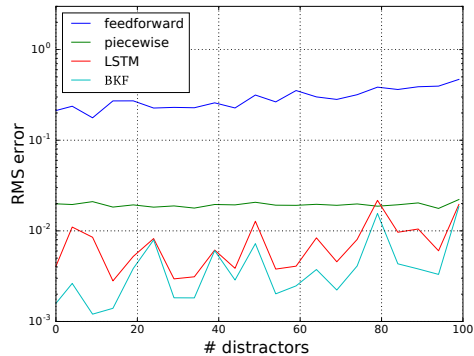

Figure 5: The RMS error of various models trained on a single training set that contained sequences of varying difficulty. The models were then evaluated on several test sets of fixed difficulty.

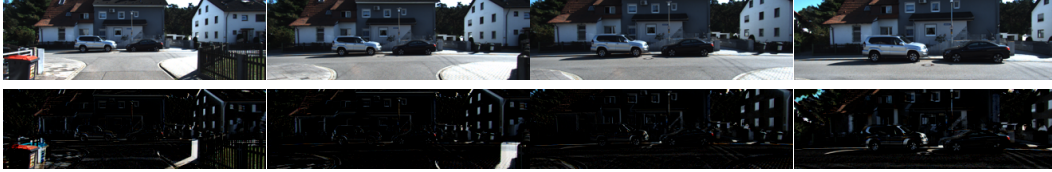

Figure 6: Example image sequence from the KITTI dataset (top row) and the corresponding difference image that is obtained by subtracting the RGB values of the previous image from the current image (bottom row). The observation $o_t$ is formed by concatenating the two images into a six-channel feature map which is then treated as an input to a convolutional neural network. The figure shows every fifth sample from the original sequence for illustrative purpose.

Table 2: KITTI Visual Odometry Results

|  | Test 100 | | | Test 100/200/400/800 | | |
|---|---|---|---|---|---|---|
| # training trajectories | 3 | 6 | 10 | 3 | 6 | 10 |
| Translational Error [m/m] | | | | | | |
| piecewise KF | 0.3257 | 0.2452 | 0.2265 | 0.3277 | 0.2313 | 0.2197 |
| LSTM model (128 units) | 0.5022 | 0.3456 | 0.2769 | 0.5491 | 0.4732 | 0.4352 |
| LSTM model (256 units) | 0.5199 | 0.3172 | 0.2630 | 0.5439 | 0.4506 | 0.4228 |
| **BKF (ours)** | **0.3089** | **0.2346** | **0.2062** | **0.2982** | **0.2031** | **0.1804** |
| Rotational Error [deg/m] | | | | | | |
| piecewise KF | 0.1408 | 0.1028 | 0.0978 | 0.1069 | 0.0768 | 0.0754 |
| LSTM model (128 units) | 0.5484 | 0.3681 | 0.3767 | 0.4123 | 0. 3573 | 0.3530 |
| LSTM model (256 units) | 0.4960 | 0.3391 | 0.2933 | 0.3845 | 0.3566 | 0.3221 |
| **BKF (ours)** | **0.1207** | **0.0901** | **0.0801** | **0.0888** | **0.0587** | **0.0556** |

## 5.4 KITTI Visual Odometry Experiment

Next, we evaluated the state estimation models on visual odometry task in the KITTI dataset [8] (Figure 6, top row). The publicly available training set contains 11 trajectories of ego-centric video sequences of a passenger car driving in suburban scenes, along with ground truth position and orientation. The dataset is challenging since it is relatively small for learning-based algorithms, and the trajectories are visually very diverse. For training the Kalman filter variants, we used a simplified state-space model with three of the state variables corresponding to the vehicle's 2D pose (two spatial coordinates and heading) and two for the forward and angular velocities. Because the dynamics model is non-linear, we equipped our model-based state estimators with extended Kalman filters, which is a straightforward addition to the BKF framework.

The objective of the task is to estimate the relative change in the pose during fixed-length subsequences. However, because inferring the pose requires integration over all past observations, a simple feedforward model cannot be used directly. Instead, we trained a feedforward network, consisting of four convolutional and two fully connected layers and having approximately half a million parameters, to estimate the velocities from pairs of images at consecutive time steps. In practice, we found it better to use a difference image, corresponding to the change in the pixel intensities between the images, along with the current image as an input to the feedforward network (Figure 6). The ground truth velocities, which were used to train the piecewise KF as well as to pretrain the other models, were computed by finite differencing from the ground truth positions. The recurrent models–piecewise KF, the BKF, and the LSTM model–were then fine-tuned to predict the vehicle's pose. Additionally, for the LSTM model, we found it crucial to pretrain the recurrent layer to predict the pose from the velocities before fine-tuning.

We evaluated each model using 11-fold cross-validation, and we report the average errors of the held-out trajectories over the folds. We trained the models by randomly sampling subsequences of 100 time steps. For each fold, we constructed two test sets using the held-out trajectory: the first set contains all possible subsequences of 100 time steps, and the second all subsequences of lengths 100, 200, 400, and 800.[1] We repeated each experiment using 3, 6, or all 10 of the sequences in each training fold to evaluate the resilience of each method to overfitting.

Table 2 lists the cross-validation results. As expected, the error decreases consistently as the number of training sequences becomes larger. In each case, BKF outperforms the other variants in both predicting the position and heading of the vehicle. Because both the piecewise KF and the BKF incorporate domain knowledge, they are more data-efficient. Indeed, the performance of the LSTM degrades faster as the number of training sequences is decreased. Although the models were trained on subsequences of 100 time steps, they were also tested on a set containing a mixture of different sequence lengths. The LSTM model generally failed to generalize to longer sequences, while the Kalman filter variants perform slightly better on mixed sequence lengths.

## 6 Discussion

In this paper, we proposed a discriminative approach to state estimation that consists of reformulating probabilistic generative state estimation as a deterministic computation graph. This makes it possible to train our method end-to-end using simple backpropagation through time (BPTT) methods, analogously to a recurrent neural network. In our evaluation, we present an instance of this approach that we refer to as the backprop KF (BKF), which corresponds to a (extended) Kalman filter combined with a feedforward convolutional neural network that processes raw image observations. Our approach to state estimation has two key benefits. First, we avoid the need to construct generative state space models over complex, high-dimensional observation spaces such as raw images. Second, by reformulating the probabilistic state-estimator as a deterministic computation graph, we can apply simple and effective backpropagation and stochastic gradient descent optimization methods to learn the model parameters. This avoids the usual challenges associated with inference in continuous, nonlinear conditional probabilistic models, while still preserving the same representational power as the corresponding approximate probabilistic inference method, which in our experiments corresponds to approximate Gaussian posteriors in a Kalman filter.

Our approach also can be viewed as an application of ideas from probabilistic state-space models to the design of recurrent neural networks. Since we optimize the state estimator as a deterministic computation graph, it corresponds to a particular type of deterministic neural network model. However, the architecture of this neural network is informed by principled and well-motivated probabilistic filtering models, which provides us with a natural avenue for incorporating domain knowledge into the system.

Our experimental results indicate that end-to-end training of a discriminative state estimators can improve their performance substantially when compared to a standard piecewise approach, where a discriminative model is trained to process the raw observations and produce intermediate low-dimensional observations that can then be integrated into a standard generative filter. The results also indicate that, although the accuracy of the BKF can be matched by a recurrent LSTM network with a large number of hidden units, BKF outperforms the general-purpose LSTM when the dataset is limited in size. This is due to the fact that BKF incorporates domain knowledge about the structure of probabilistic filters into the network architecture, providing it with a better inductive bias when the training data is limited, which is the case in many real-world robotic applications.

In our experiments, we primarily focused on models based on the Kalman filter. However, our approach to state estimation can equally well be applied to other probabilistic filters for which the update equations (approximate or exact) can be written in closed form, including the information filter, the unscented Kalman filter, and the particle filter, as well as deterministic filters such as state observers or moving average processes. As long as the filter can be expressed as a differentiable mapping from the observation and previous state to the new state, we can construct and differentiate the corresponding computation graph. An interesting direction for future work is to extend discriminative state-estimators with complex nonlinear dynamics and larger latent state. For example, one could explore the continuum of models that span the space between simple KF-style state estimators and fully general recurrent networks. The trade-off between these two extremes is between generality and domain knowledge, and striking the right balance for a given problem could produce substantially improved results even with relative modest amounts of training data.

**Acknowledgments**

This research was funded in part by ONR through a Young Investigator Program award, by the Army Research Office through the MAST program, and by the Berkeley DeepDrive Center.

## Footnotes

[1]The second test set aims to mimic the official (publicly unavailable) test protocol. Note, however, that because the methods are not tested on the same sequences as the official test set, they are not directly comparable to results on the official KITTI benchmark.

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
