[Supplementary Material]

# A   Backprop KF Computation Graph Architecture

The architecture of the computation graph corresponding to the BKF used in the synthetic tracking task is shown in Figure 3. It consists of a feedforward neural network (blue blocks); recurrent part based on Kalman filter (red blocks); and a block that guarantees positive semidefiniteness of the observation covariance as well as the loss function (gray blocks). The network outputs a point estimate of the intermediate observation $\mathbf{z}_t$ and a vector $\hat{\mathbf{L}}_t$, which is used to estimate the covariance $\mathbf{R}_t$ of $\mathbf{z}_t$. We enforce the positive definiteness of $\mathbf{R}_t$ through Cholesky decomposition by constructing a lower-triangular matrix $\mathbf{L}_t$ from $\hat{\mathbf{L}}_t$ with exponentiated diagonal elements and by setting $\mathbf{R}_t = \mathbf{L}_t \mathbf{L}_t^{\mathsf{T}}$. The full description of network is given in Table 3.

The Kalman filter is based on the following dynamical system:

$$
\begin{aligned}
\mathbf{x}_{t+1} &= f(\mathbf{x}_t) + \mathbf{B_w} \mathbf{w}_t \quad \text{\textit{dynamics update}} \\
\mathbf{z}_t &= \mathbf{C_z} \mathbf{x}_t + \mathbf{v}_t \qquad \text{\textit{observation inferred from } } \mathbf{o}_t : \ \mathbf{z}_t = g_\theta(\mathbf{o}_t) \\
\mathbf{y}_t &= \mathbf{C_y} \mathbf{x}_t \qquad\qquad \text{\textit{ground truth observation}}
\end{aligned}
\tag{4}
$$

The state vector $\mathbf{x}_t$ tracks the position and velocity of the system. In the case of linear dynamics, the dynamics function reduces to $f(\mathbf{x}_t) = \mathbf{A}\mathbf{x}_t$. The dynamics noise, $\mathbf{w}_t$, and the observation noise, $\mathbf{v}_t$, are assumed to be IID zero mean Gaussian random variables with covariances $\mathbf{Q}$ and $\mathbf{R}_t = \mathbf{L}_t \mathbf{L}_t^{\mathsf{T}}$, respectively.

The Kalman filter updates are given by

$$
\begin{aligned}
\boldsymbol{\mu}'_{\mathbf{x}_{t+1}} &= \mathbf{A}\boldsymbol{\mu}_t & \text{\textit{mean dynamics update}} \\
\boldsymbol{\Sigma}'_{\mathbf{x}_{t+1}} &= \mathbf{A}\boldsymbol{\Sigma}_{\mathbf{x}_t}\mathbf{A}^{\mathsf{T}} + \mathbf{B_w}\mathbf{Q}\mathbf{B_w}^{\mathsf{T}} & \text{\textit{covariance dynamics update}} \\
\mathbf{K}_{t+1} &= \boldsymbol{\Sigma}'_{\mathbf{x}_{t+1}}\mathbf{C}^{\mathsf{T}}(\mathbf{C}\boldsymbol{\Sigma}'_{\mathbf{x}_{t+1}}\mathbf{C}^{\mathsf{T}} + \mathbf{R}_{t+1})^{-1} & \text{\textit{Kalman gain}} \\
\boldsymbol{\mu}_{\mathbf{x}_{t+1}} &= \boldsymbol{\mu}'_{\mathbf{x}_{t+1}} + \mathbf{K}_{t+1}(\mathbf{z}_{t+1} - \mathbf{C}\boldsymbol{\mu}'_{\mathbf{x}_{t+1}}) & \text{\textit{mean observation update}} \\
\boldsymbol{\Sigma}_{\mathbf{x}_{t+1}} &= (\mathbf{I} - \mathbf{K}_{t+1}\mathbf{C})\boldsymbol{\Sigma}'_{\mathbf{x}_{t+1}} & \text{\textit{covariance observation update}}.
\end{aligned}
\tag{5}
$$

The extended version of Kalman filters is obtained by setting $\mathbf{A} = \dfrac{\partial f\left(\boldsymbol{\mu}'_{\mathbf{x}_{t+1}}\right)}{\partial \mathbf{x}_{t+1}}$. The recurrence begins with the observation update. The covariance $\boldsymbol{\Sigma}'_{\mathbf{x}_0}$ is considered as a hyper parameter, and the initialization scheme for the mean $\boldsymbol{\mu}'_{\mathbf{x}_0}$ depends on the task at hand. Note that the labels $\mathbf{y}_t$ represent noiseless observations and are not incorporated in the Kalman filter updates, but instead they enter the model at training time through the cost function

$$
l(\theta) = \sum_{i=1}^{N}\sum_{t=1}^{T} \frac{1}{2TN}(\mathbf{C_y}\boldsymbol{\mu}_{\mathbf{x}_t}^{(i)} - \mathbf{y}_t^{(i)})^{\mathsf{T}}(\mathbf{C_y}\boldsymbol{\mu}_{\mathbf{x}_t}^{(i)} - \mathbf{y}_t^{(i)}),
\tag{6}
$$

where $N$ and $T$ denote the number and the length of training sequences, respectively.

# B   Synthetic Tracking Experiment

In the tracking experiment, the state vector $\mathbf{x}_t$ corresponds to the 2D position and the velocity of the red disk. The dynamics model is a simple integrator, and the dynamics noise is applied only to the velocity of the disk.

The feedforward network used in the experiments is detailed in Table 3. The network gets a third person view image as an observation $\mathbf{o}_t$ (Figure 4). The filter state variables $\boldsymbol{\mu}'_{\mathbf{x}_0}$ and $\boldsymbol{\Sigma}'_{\mathbf{x}_0}$ are initialized with the ground truth state identity matrix. We chose this initialization scheme to suppress the initial transient that would result if the initial estimation error was large.

We trained the model with the ADAM [13] optimizer with the default decay rates. We manually picked a learning rate that resulted into the lowest validation error and it varied among the models. We first trained the feedforward net to predict the position of the red disk (feedforward model). The same parameters where then used in the piecewise KF that filters the feedforward predictions and learns a constant observation covariance $\mathbf{R}$. The feedforward model was then further fine-tuned to predict also $\mathbf{R}_t$ with a maximum-likelihood objective. The resulting parameters were used to

Table 3: Feedforward networks

| Tracking | Visual Odometry |
|---|---|
| 128x128x3 input tensor, image | 150x50x6 input tensor, image + difference image |
| 9x9 conv, 4, stride 2x2; ReLU; resp norm | 7x7 conv, 16, stride 1x1; ReLU; resp norm |
| max-pool 2x2, stride 2x2 | 5x5 conv, 16, stride 2x1; ReLU; resp norm |
| 9x9 conv, 8, stride 2x2; ReLU; resp norm | 5x5 conv, 16, stride 2x1; ReLU; resp norm |
| max-pool 2x2, stride 2x2 | 5x5 conv, 16, stride 2x2; ReLU; resp norm; dropout, 0.9 |
| fc, 16, ReLU | fc, 128, ReLU |
| fc, 32, ReLU | fc, 128, ReLU |
| fc, 2 (output, $\mathbf{z}_t$); fc, 3 (output, $\hat{\mathbf{L}}_t$) | fc, 2 (output, $\mathbf{z}_t$); fc, 3 (output, $\hat{\mathbf{L}}_t$) |

initialized both the BKF and the LSTM models. In the case of LSTM, we also removed the output layers of the feedforward network, and concatenated the last hidden layer activations with the ground truth initial state of the disk.

## C   KITTI Visual Odometry

In the 2D visual odometry experiment, the state vector $\mathbf{x}_t$ is comprised of the coordinates and the heading of the vehicle, expressed in the inertial frame, as well as forward and angular velocities, expressed in the ego-centric frame, resulting into five state variables. The observation $\mathbf{z}_t$ corresponds to the velocities, and the ground truth observation $\mathbf{y}_t$ to the position and the heading. The resulting non-linear dynamics model is then linearized at the current state estimate in accordance with standard extended Kalman filter updates.

The feedforward network architecture for visual odometry is listed in Table 3. We followed the same training procedure as explained in Appendix B, with an exception that the feedforward network was pretrained to predict the velocities, and we omitted the additional pretraining step on the maximum-likelihood objective. Moreover, for the Kalman filter variants, we learned additionally the dynamics covariance. For the LSTM model, we found it to be crucial to pretrain also the recurrent layers to predict $\mathbf{y}_t$ directly from the ground truth velocities. Therefore, we did not eliminate the bottleneck output layers of the feedforward network during the final end-to-end finetuning of the LSTM model. Although we used only monocular camera as an input, the dataset has stereo observation which we treated as two independent monocular sequences. We further augmented the dataset with mirror images by flipping them vertically and reversing the sign of the ground truth heading.