[Reviews · NeurIPS 2016]

Reviewer 1

Summary

The paper describes that the standard Kalman Filter (KF) equations can be expressed in a computation graph, thereby integrating probabilistic state estimation with a per-frame convnet. This enables optimizing the convnet and the resulting low-dimensional features with respect to the filtered state directly on a batch of images with known target location using standard back-propagation. The hypothesis is that a KF provides useful domain knowledge for state estimation (i.e. through the linear motion equations and the parametric state probability distribution) over a generic LSTM. The trained network yields thereby a better representation for state estimation of the target in novel tracking sequences, as compared to training the convnet on individual frames independently without considering the KF. Also, the convnet uses a novel normalization layer which resembles batch normalization. The method is evaluated on an artificial tracking problem consisting of a red disc moving through and occluded by various other colored discs, and compared to independent training of the convnet, and a version where the KF is replaced by LSTM. The results show that the proposed method outperforms these baselines on the artificial dataset.

Qualitative Assessment

I like the idea to investigate integration of the well-known Kalman Filtering in back-propogation, to obtain best of both-worlds: data-driven feature learning and informed motion models. The paper is generally well written and easy to follow. Since one could expect learning motion models would also benefit from non-linear data-driven models (e.g. LSTM) with large amounts of data, instead of using highly informed models with few free parameters, the question when/where this trade-off occurs would be very interesting. The idea that this could improve state estimation in tracking specifically is also appealing, especially if empirical evidence on real-world tracking tasks can be given. However, this paper does not really explore the first question. It is unclear how much data is used, and if the exhibited movements are realistic or already very similar towards the linear equations used in the Kalman filter. Also, the presented tracking use-case seems very ad-hoc and not very relevant to any real-world tracking use-case: it presuposes that we posses extensive training data of the instance that we wish to track (a red circle) with a distinct and static appearance, sufficient to train a convnet for the specific appearance. Standard model-based tracking (e.g. multi-person tracking) or model-free tracking (e.g. track unknown object from initial bounding box) do not fit this task, as the instance appearances are not known a-priori. So while the artificial scenario is good for illustration and explaining the method, it does not really "reflect typical challenges in visual state estimation" (line 202). Due to the limited experimental evaluation and unclear use-case, I rated both Technical Quality and the Potential Impact as substandard for NIPS. More details comments follow below. ## Detailed comments: - line 104: In this example, what would the hidden system state x_t be? If the state represents the 'true' position and velocity, do you consider y_t a noisy version of x_t (if yes, then why consider y_t the output instead of x_t, if no why call it an observation in a Kalman Filter)? - line 181. Should L not be defined as the negative log likelihood? I.e. higher likelihood = lower loss. - line 136: Aren't the two terms in the brackets ("dphi/ds . dL/dphi" and "dL / ds") the same? I would expect the recursion to take the form of "dL / ds_{t-1} = dL / dphi_{t-1} . dphi_{t-1} / ds_{t-1} + ds/ds_{t-1} . dL/ds" - line 183. introduces a new contribution, namely a novel response normalization layer. Such a novelty warrants extra experiments (e.g. comparing to batch normalization [9]), or at least some insight into why it is "substantially more effective" (line 188). Can you expand on/cite why standard approach "is know(n) to require additional care"? - line 195.5: Are the KF parameters also trained using back propagation, e.g. Q, A, B, C ? If not, how are they determined? - The experimental section provides no details on how much training, testing data is used, what kind of back-propogation technique, or any other details to reproduce results. Are the results averaged over multiple folds / new batches of artificial data? For instance, are the up-down patterns in Fig. 6b systematic or due to random effects in a single artificial sequence? - In the presented artificial dataset, the target seems to have a clearly distinct and static appearance (its color) compared to almost all distractors. So, while technically the model is processing "raw pixel data", this seems to overstate the complexity of dealing with real-world raw pixel data a lot: Creating a good per-frame target response map appears trivial (a simple color threshold), which undermines the whole point of using a convnet to learn a robust representation. - The main question therefore remains: does this approach help to improve state estimation in real tracking scenarios where the target is not known a-prior, where the appearance is dynamic, has lighting and shape changes, and fore and background as more prone to false positives? E.g. comparing tracking on VOT 2014/2015 sequences, or PETS pedestrian sequences would be more convincing. Also, in normal tracking scenarios you do not have labeled prior examples of the specific target *instance*. At best, you have labeled examples of the target class (e.g. pedestrians). A tracker still needs to online learn a way to discriminate other instances of the same class. For instance, could the method be used to improve and initial tracking results from a baseline tracking method providing inaccurate object labels? ## Minor comments - Equations are not numbered, annoying for anyone referring to parts of the paper. - line 138.5: At this point I had several questions (though these were somewhat clarified later in Section 5.2): How are these gradient terms obtained in practice? Why not provide the closed form solutions, or appropriate citations? Or do the authors rely on automatic derivation by a software package? Is there any risk of a gradient descent step leading to non positive-definite covariance? - line 169: Extended Kalman Filter should have a citation, it was not introduced anywhere before. - line 261.: "although the accuracy of the BKF can be matched by a recurrent LSTM network". Where is this shown in the experiments? How could one conclude this from Table 1, or Figure 6?

Confidence in this Review

2-Confident (read it all; understood it all reasonably well)


Reviewer 2

Summary

This paper combines a convolutional neural network with a Kalman filter in order to model a generative dynamical system without having to explicitly create a generative model of images. Essentially, one can train a CNN on images to predict some target, e.g., the coordinates of an object, and then use the representation as an input to a Kalman filter. The approach in this paper is to instead treat this process as one single computational graph, and to backpopagate through the Kalman filter to tune everything together.

Qualitative Assessment

I like the idea in this paper as I think combining probabilistic models and neural networks is an interesting avenue of research. The paper itself is well written, although the description of the model is a bit scattered and there are some points where I think things could be more clear (I will elaborate below). To be direct though, I think that the main thing this paper lacks are a comprehensive set of experiments. The paper discusses applications in robotics and automobiles, but then only demonstrates the proposed model on a toy dataset. I think that this task is challenging due to the presence of clutter and occlusion, however it's unclear where the win is coming from compared to the baselines. Is it coming from fewer parameters, so less overfitting? Is it coming from the linear dynamics of the dataset? Does the explicit characterization of uncertainty allow the model to dynamically alter the weighting that trades off the system dynamics and the observations? I would like clarification on this, but it seems like the dynamics of the dataset are probably linear, which makes the Kalman filter a good model of this system. I would expect that many real-world systems are not linear, in which case I would like to see how robust this approach is, and whether it can still outperform an LSTM. I understand that this approach can be extended to nonlinear dynamics, but this is only discussed and never demonstrated (and the authors will probably argue that it's beyond the scope of this paper). Basically, I think that the experiment is promising, but I would like to see either a more thorough empirical investigation, or a more compelling application. In terms of clarity, I didn't find the description of the model in terms of the derivatives of the system very helpful. I'm guessing that anyone implementing this would simply use an automatic differentiation framework. Also maybe I'm interpreting this incorrectly, but it seems like there is an error in the formula just below line 136? I think that the Kalman filter in Figure 3 should map more explicitly onto the model described in section 5.2. Namely, the C parameter in Figure 3 should be labelled with Cp or Cy and the mapping from the internal state to the observations p and y should also be included. Basically, any computation done in the forward pass of the model should be included there. So to summarize: I think the idea is really intriguing and I like the work overall, but I think there needs to be at least one more convincing experiment.

Confidence in this Review

2-Confident (read it all; understood it all reasonably well)


Reviewer 3

Summary

The paper proposes a method for discriminative training of state estimators by constructing deterministic computation graph and optimizing it using gradient descent methods and backpropagation techniques from recurrent neural networks. In particular, the authors propose a method for designing neural network architectures with close applications in state estimation. One of the new aspects of this method is its capability in training of expressive discriminative models and its applicability in handling nonlinear features into the observation and transition functions. The idea in this method is to view the filter as a computation graph unrolled through time with internal states being the posteriors over hidden states and training a discriminativ model on the entire sequence of observations rather than just single time steps. The paper also provides some experimental results to compare the performance of an instance of the above method (which is a combination of Kalman filter with feedforward convolutional neural network) with some other techniques such as standard Kalman filter.

Qualitative Assessment

The introduction can be better motivated by providing some explicit real world applications and mentioning that how much the proposed model can effectively improve the performance of the earlier methods. In page 2, last paragraph, please elaborate further on the main differenced of your proposed method and those given in references [3,18]. The definitions and concepts given in sections 3 and 4 are quite informal. Since the main method is defined in these parts, it is worth to describe them more clearly. The test beds and data sets in section 5 are not precisely defined which make it hard to conclude a fair evaluation for the performance of the proposed method. In most parts, the authors only provide a short summary of the results which at some cases it is even hard to the readers to regenerate the same results. So it would be better to provide more details for the experimental results and instead remove some of the extra discussions from section 6.

Confidence in this Review

1-Less confident (might not have understood significant parts)


Reviewer 4

Summary

The paper describes an end-to-end trained state estimator for high-dimensional input, combining a CNN (in case of visual input), and a recurrent neural network / kalman filter so that all weights in the resulting network can be trained using back propagation / bptt 

Qualitative Assessment

Technical quality is OK though it would have been good to see how different input modalities can be handled. The relation between RNN and kalman filter has also been explored before, in eg. http://papers.nips.cc/paper/3665-a-neural-implementation-of-the-kalman-filter.pdf It would have been interesting to learn more about the learned intermediate representation; in what way is it different from separately learning state (and filter). Also the specific dynamics of the model may just be well suited for the solution but it's hard to say from the paper and just one scenario.

Confidence in this Review

2-Confident (read it all; understood it all reasonably well)


Reviewer 5

Summary

The paper proposed a recurrent structure to train state estimators in an end-to-end way using gradient descent. The proposed method is mainly designed for state estimators whose inputs are complex (e.g., raw camera images). In experiment section, the authors proposed a specific structure: Backprop Kalman Filter, designed for learning state estimator for linear dynamical systems. Suggestions about how to design the structure for learning state estimators of non-linear systems are briefly mentioned in the discussion section.

Qualitative Assessment

1. Related work: one advantage of the proposed method is that it directly optimize the state estimator with respect to the prediction error (discriminative loss). This concept is very similar to some previous works such as Inference Machine (Langford et.al, 2009 ICML, Ross et.al, 2011 CVPR, Sun et.al, 2016 ICML) and the recently proposed method Deep Tracking at AAAI 2016 (Ondruska & Posner, 2016 AAAI). I think the concept and the structure proposed in this paper is quite similar to Deep Tracking. 2. In the experiment, the proposed backdrop KF structure is limited to linear systems. It is not surprising that backprop KF beats LSTM in the experiment since the experiment uses a linear dynamical systems. I would guess it is possible that LSTM can beat backdrop KF if for non-linear dynamical systems. Though the authors mentioned in the discussion section that it is possible to extend KF to non-linear state estimators such as UKF and particle filter, more details are needed for how to design the whole structure and perhaps corresponding experiments are necessary to convince readers that it can be properly extended to non-linear dynamical systems. Otherwise, I think the contribution of the paper is only limited to learning state estimators for linear dynamical systems. I think for general non-linear dynamical systems, it is non-trivial to incorporate prior knowledge of the system's form into the design of the computational graph. In these cases, general purpose recurrent structure such LSTM and Deep Tracking may perform better. 3. For the piecewise KF model, it is unclear how the feedforward neural and filters are trained separated. Especially for training the feedforward neural network, is there training data of p_t? Also what's the details of the CNN used for this baseline? Does it share the similar structure of the one used in BKF? Also since the paper considers latent state space models, for the simple "Feedforward Model" baseline, one probably should at least try to compare to Autoregressive model: namely feed a history of observations {o_{t-k},...,o_t} to CNN instead of a single observation. 3. In Fig 3, It is unclear where A, C, and C_f are from. Are they treated as parameters and learnt by back propagation as well?

Confidence in this Review

2-Confident (read it all; understood it all reasonably well)


Reviewer 6

Summary

The paper introduces a general framework for designing and learning discriminative state estimators framed as computational graphs, that can handle high dimensional observations and can be learned efficiently in an end-to-end fashion to directly optimize the outcome of the state estimator. One realization of the framework consisting of a convolutional neural network and a Kalman filter is evaluated on a visual tracking task and compared to recent competing models.

Qualitative Assessment

Generally the paper is well written and structured, the model is explained understandingly and the results are analyzed well. If the presented results can be achieved at other tasks as well to validate the models strength, the presented work has the potential to be useful in many different areas, especially as the introduced framework is quite general. The explanation of the backpropagation algorithm, especially the recursive equation, could be more detailed. As the authors introduce their framework as a general framework, it would be beneficial for validating that statement if they would include another realization in the results. The presented realization consisting of a convolutional neural network and Kalman filter as a recurrent part is well explained and illustrated. Although, the learning scheme for that model and parameters are missing, such that it cannot be re-implemented to either validate the results or to compare to it. Another detail of the CNN is missing; as the authors use a 'novel response normalization', a more detailed description of it would be advisable. One additional experiment with a different task should be considered by authors to confirm the amazing result compared to the competing models. Suggestion about missing related reference: Rahul G. Krishnan, Uri Shalit, David Sontag, Deep Kalman Filters, http://arxiv.org/abs/1511.05121 They use a generative model based on deep neural networks trained with backpropagation and applied it to images as well. The two works differ, but they are related and I would suggest to include it into the related work and, if manageable, compare to it in the experiment as well.

Confidence in this Review

2-Confident (read it all; understood it all reasonably well)